# Real-Time Measurement of CH_4_ in Human Breath Using a Compact CH_4_/CO_2_ Sensor

**DOI:** 10.3390/s24041077

**Published:** 2024-02-07

**Authors:** Yueyu Lin, Dexter Manalili, Amir Khodabakhsh, Simona M. Cristescu

**Affiliations:** Life Science Trace Detection Laboratory, Department of Analytical Chemistry and Chemometrics, Institute for Molecules and Materials, Radboud University, 6525 AJ Nijmegen, The Netherlands; yueyu.lin@ru.nl (Y.L.); dexter.manalili@gmail.com (D.M.); amir.khodabakhsh@ru.nl (A.K.)

**Keywords:** methane, carbon dioxide, breath, correction functions, real-time measurements

## Abstract

The presence of an elevated amount of methane (CH_4_) in exhaled breath can be used as a non-invasive tool to monitor certain health conditions. A compact, inexpensive and transportable CH_4_ sensor is thus very interesting for this purpose. In addition, if the sensor is also able to simultaneously measure carbon dioxide (CO_2_), one can extract the end-tidal concentration of exhaled CH_4_. Here, we report on such a sensor based on a commercial detection module using tunable diode laser absorption spectroscopy. It was found that the measured CH_4_/CO_2_ values exhibit a strong interference with water vapor. Therefore, correction functions were experimentally identified and validated for both CO_2_ and CH_4_. A custom-built breath sampler was developed and tested with the sensor for real-time measurements of CH_4_ and CO_2_ in exhaled breath. As a result, the breath sensor demonstrated the capability of accurately measuring the exhaled CH_4_ and CO_2_ profiles in real-time. We obtained minimum detection limits of ~80 ppbv for CH_4_ and ~700 ppmv for CO_2_ in 1.5 s measurement time.

## 1. Introduction

Methane (CH_4_) is an end-product of microbial fermentation of the undigested polysaccharides fraction of carbohydrates in the human gut [1,2]. Its accumulation within the intestine may result in abdominal bloating or distension [3]. Methane is absorbed into the bloodstream and excreted in flatus and breath [4]. Hydrogen and CH_4_ breath tests are nowadays used to assess for carbohydrate malabsorption (e.g., lactose or fructose intolerance) [5] or small bowel bacterial overgrowth [6,7]. Recently, CH_4_ has been found to play a role in gastrointestinal inflammatory diseases and colorectal cancer [8,9]. 

Breath analysis for medical diagnosis relies on collecting the end-tidal fraction of the exhaled air [10]. This last part of exhaled air reflects best the composition of the alveolar air and can be identified by monitoring the expired carbon dioxide (CO_2_). Thus, recording exhaled CO_2_ at the same time with CH_4_ allows for the measurement of end-tidal CH_4_ concentrations [11].

Different techniques, including spectroscopic, electrochemical, calorimetric, solid-state and piezoelectric methods, as well as gas chromatography (GC) equipped with a flame-ionization detector (FID) and real-time mass spectrometry, have been applied to monitor CH_4_ levels in human breath [3,5,8,9,12,13,14,15]. There are also commercial instruments available, often in combination with an electrochemical sensor for hydrogen [3]. Inexpensive methods, such as electrochemical, piezoelectric and calorimetric detection systems, are usually not very selective and are quite sensitive to environmental changes, such as humidity and temperature. Gas chromatography and real-time mass spectrometry are rather large and expensive [16,17].

Several spectroscopic detection techniques have been employed for measuring exhaled CH_4_ [18]. Sensors based on photoacoustic spectroscopy have shown a high capability for monitoring the exhaled CH_4_ in humans [19,20] and animals [21,22]; however, they usually need to be regularly calibrated. Methane sensors based on cavity ring-down spectroscopy have also used for this purpose, providing a calibration-free and highly sensitive detection [23], but they are rather expensive. Considering the available instruments, there is an increased interest in developing competitive, calibration-free, real-time and low-cost CH_4_ sensors using tunable diode lasers in the near-infrared (NIR) region. These light sources in combination with a small-volume multi-pass cell can be integrated into a compact design OEM module with easy operation as the detection part of a gas sensor [24,25,26,27,28].

Here we report on the development of a portable sensor based on a commercial OEM (original equipment manufacturer) module (LGD Compact-A CH_4_/CO_2_, Axetris, Kaegiswil, Switzerland) for real-time measurements of CH_4_ and CO_2_ concentrations in exhaled human breath. For simplicity, we will refer to the commercial OEM module as the “LGD module”, in which LGD stands for “laser gas detection”. Since breath is a complex mixture of volatiles and has a high relative humidity, our aim is to identify whether the CH_4_ and CO_2_ concentrations provided by the LGD module are influenced by water vapor and eventually apply the necessary corrections. Afterwards, we demonstrate the performance of the developed sensor for real-time end-tidal measurement of CH_4_/CO_2_ in the exhaled breath of few participants. We also study the temperature dependence, detection sensitivity and long-term stability of the sensor to provide a full picture of the capabilities of the system.

## 2. Materials and Methods

### 2.1. Experimental Set-Up for Investigating the Relative Humidity Influence

Our breath sensor is based on an LGD Compact-A CH_4_/CO_2_ OEM module (Axetris, Kaegiswil, Switzerland) using tunable diode laser absorption spectroscopy (TDLAS) operating in the NIR region and a small Herriott multipass cell (19 mL). A single tunable diode laser scans over an absorption line of each target gas (CH_4_ and CO_2_) and the transmitted light is recorded by a photodetector. The absorbance of each gas is calculated according to the Beer–Lambert law and the concentration of the gases is retrieved using the optical path length of the multipass cell and the absorption coefficient of the target gas. The possible detection interference between CH_4_ and CO_2_ is addressed and compensated by the manufacturer. The specific NIR wavelength range used in the module is not specified by the vendor due to the protection of the commercial product from competition; however, the absorption spectrum of CH_4_ within the NIR region is heavily overlapped by that of water, as can be easily checked using the HITRAN database [29]. The magnitude of this spectral interference is drastic in breath analysis, since the exhaled breath is almost saturated with water vapor.

To investigate this interference and correct for the effects of humidity on the CH_4_ measurements, we developed an experimental setup, as depicted in Figure 1. It includes the LGD module and a highly accurate commercial gas analyzer (Ultra-portable Greenhouse Gas Analyzer 915-0011, Los Gatos Research, Mountain View, CA, USA) used as the reference instrument. This gas analyzer is based on off-axis integrated cavity output spectroscopy (OA-ICOS) and provides very high sensitivity and accuracy, but it is several times more expensive than the LGD module. Two cylinders with certified mixtures of CH_4_ in synthetic air (80% nitrogen and 20% oxygen) of 101.5 (±2) ppmv and of 9.59 (±0.2) ppmv, respectively, and one cylinder with certified mixture of CO_2_ in synthetic air of 4.96 (±0.1)% are used for this experiment (all from Linde Gas Benelux, Dieren, The Netherlands). The output of the bottles are further diluted with pure nitrogen, using two mass flow controllers (MFC) (EL-FLOW Prestige, Bronkhorst, AK Ruurlo, The Netherlands), to yield different concentrations of CH_4_/CO_2_ in nitrogen/synthetic air. 

The gas sample is humidified by passing through the headspace of a closed vial (500 mL volume) partially filled with water, where the relative humidity is controlled by adjusting the amount of water in the vial (via an attached syringe). The humid gas flow is further transported to the multipass cell of the LGD module and afterwards to the reference analyzer. The relative humidity of gas samples is measured by a relative humidity sensor (HygroPalm HP32, ROTRONIC AG, Bassersdorf, Switzerland) placed after the LGD module and before the reference analyzer. The total flow rate is set at 11 mL/s and a vacuum pump in the reference analyzer maintains constant atmospheric pressure in the cell of the reference analyzer. Data acquisition is achieved with a regular PC (Windows 7, 8 GB memory, Intel(R) Core (TM) i5-3470 CPU @3.20 GHz).

### 2.2. Experimental Setup for On-Line Breath Measurement

To evaluate the performance of the LGD module in analyzing CH_4_ in human breath, a custom-built breath sampling setup is designed and implemented. The participant exhales freely through an HME filter (9066/701, GVS Filter Technology, Zola Predosa, Italy) and a CAPNOSTAT^®^ 5 CO_2_ sensor (Respironics Novametrix, Wallingford, CT, USA) that is continued with a one-way breathing valve. The outlet of the valve is connected to a buffer tube of 10 mL volume provided with a small aperture to draw a constant flow of the breath sample (33 mL/s) into the LGD multipass cell using a small membrane pump (N86KT.18, KNF, Vleuten, The Netherlands). The buffer volume is needed for extending the time to analyze the end-tidal breath. Figure 2 shows the schematic representation of the setup for on-line breath measurements.

The CAPNOSTAT^®^ 5 CO_2_ sensor monitors the mainstream of exhaled CO_2_ and is connected to a commercial breath sampler (Loccioni, Angeli di Rosora, Italy) that allows the exhalation profile to be displayed on a screen. It also serves as the reference for CO_2_ measurement with the LGD module-based sensor in case of real-time breath experiments. The relative humidity sensor is inserted after the sensor for calibration during the data processing.

### 2.3. Sample Preparation and Data Processing

To determine if the cross-detection effects are concentration-dependent or not, we vary the concentrations of CH_4_ and CO_2_ and check the effects of humidity. By using two mass flow controllers (EL-FLOW Prestige FG-201CV, Bronkhorst, AK Ruurlo, The Netherlands), CH_4_ and CO_2_ from separate cylinders are diluted in nitrogen to produce 2, 4, 6, 8, 10, 25, 50 and 100 ppmv, and 2%, 3% and 5% concentrations, respectively. Due to the difficulty in accurately controlling the relative humidity of the gas mixture consistently, we use 5 random data points ranging from 0% to ~80% but measure these values accurately. 

In the data processing, we average 10 s for each data point. The standard deviations are calculated based on 5 consecutive measurements. For each applied concentration, a linear fitting is performed to the measured concentration values in terms of the corresponding relative humidity. The regression coefficients of the fits (slopes) are used to evaluate the sensitivity of the instrument to changes in relative humidity at different concentration levels. To investigate whether the effects of humidity are related to the applied gas concentration, we analyze the percentage changes in the gas concentrations (rather than the absolute concentration changes). After performing the fitting for each applied concentration, we identify the intercept (at 0% relative humidity), and subsequently normalize all values with respect to this intercept. By doing so, all values are transformed into percentages relative to the intercept, which should now have a numerical value of 1. Through a straightforward subtraction operation from the intercept, the percentage differences are obtained in different relative humidity levels. To calculate the standard deviation, the method of error propagation is applied.

## 3. Results and Discussion

### 3.1. Effects of Relative Humidity

#### 3.1.1. Results in Different Relative Humidity

As discussed above, different concentrations of CH_4_ and CO_2_ are applied to determine whether the cross-detection effects are concentration-dependent. Figure 3 represents the results after the re-calculation for both CH_4_ and CO_2_. For both CH_4_ and CO_2_ measured by the LGD module, the percentage changes in the values against different relative humidity levels are consistent across different concentrations. Lower concentration values have larger standard deviation due to the error propagation. Both CH_4_ and CO_2_ measured concentration decrease almost linearly by increasing the relative humidity from 0 to 80%.

#### 3.1.2. Correction Function

To account for the effects of humidity, we apply correction functions to both the CH_4_ and CO_2_ measurements form the LGD module. In addition, we calibrate all the CH_4_ values based on the values obtained from the reference analyzer at 0% relative humidity (to become independent from the accuracy of the dilution using mass flow controllers). Below are the two correction functions of CH_4_:(1)$$C_{{CH}_{4}}=\frac{sensor\;reading}{1.043-0.0434*RH}-0.384,\;\;if\;sensor\;reading>10\;ppm$$
(2)$$C_{{CH}_{4}}=\frac{sensor\;reading}{1.095-0.0402*RH}+0.013,\;\;if\;sensor\;reading<10\;ppm$$

Here $C_{{CH}_{4}}$ is the actual concentration of CH_4_ after correction, while sensor reading is the recorded data from the LGD module and RH is the relative humidity of the applied gas sample (unit: %).

For the correction of the CO_2_ concentrations, we do not use the reference analyzer since the maximum CO_2_ concentration (~5%) is beyond the operational range of the analyzer (3%). Instead, we calculate the values of CO_2_ concentrations based on diluting the output of certified CO_2_ cylinder with pure nitrogen. The correction function for CO_2_ is:(3)$$C_{{CO}_{2}}=\frac{sensor\;reading}{1-0.08*(RH-45\%)}$$

Here $C_{{CO}_{2}}$ is the actual concentration of CO_2_ after correction, while sensor reading is the recorded data from the LGD module and RH is the relative humidity of the applied sample (unit: %). 

To evaluate the performance of the correction functions, we compare the values obtained before and after correction with the values provided by the reference analyzer for CH_4_ and the diluted concentrations of the calibrated gas mixture for CO_2_. The comparison allows us to assess the accuracy of the correction functions in removing the cross-detection effect of humidity. Figure 4 shows examples for applied concentrations of ~100 ppmv CH_4_ and ~3% CO_2_ with the measurement values before and after correction obtained from the LGD module, as well as the reference values in terms of different relative humidity. After applying the respective correction functions to both CH_4_ and CO_2_, the values are much closer to the reference values. Table 1 shows the measurement errors (while varying the relative humidity from 0% to 80%) before and after correction, compared to the values from the reference analyzer for CH_4_ and the calculated concentration values after the dilution of the certified cylinder for CO_2_, for the entire range of applied concentrations. The error from the measurement is less than 1.5% (for high concentration, >10 ppmv) and less than 3% (for low concentration, <10 ppmv) for CH_4_, while the error is always below 1.4% after the correction for CO_2_. Note that our correction method can be applied to other sensors with different working principles and techniques, as long as the effect of the interfering species on the measurement stays quantifiable and reproducible.

The next step is to assess the performance of the correction functions on measured CH_4_ concentrations from the human breath. For this, we conduct an offline breath measurement. Firstly, the breath samples from three participants are collected into Tedlar bags accordingly to the protocol described in details in [30]. The same bag is measured with both the LGD module and the reference analyzer connected in series, while the relative humidity sensor is connected in between the two instruments. After correction for RH, the error of CH_4_ concentration is less than 1%, decreasing from ~6%. For CO_2_, the error is less than 1.5%, decreasing from ~3%. Therefore, it can be concluded that the correction functions can effectively eliminate the humidity interference in a breath sample, too.

### 3.2. Real-Time Breath Measurement

Finally, we record the CH_4_ and CO_2_ breath profile in real-time with the breath sensor based on the LGD module (Figure 5). Since the response time of the reference analyzer does not allow breath-to-breath monitoring, for comparison we record the CO_2_ profile from the CAPNOSTAT^®^ 5 CO_2_ sensor. The CH_4_ and CO_2_ values that are recorded by the breath sensor are shown in grey and red, respectively, while the CO_2_ values recorded by the Capnostat 5 are depicted in blue. By applying the correction functions, the CO_2_ values measured by the LGD module demonstrate a good agreement with the CAPNOSTAT^®^ 5 CO_2_ sensor for the end-tidal phase. A mismatch is observed in the CO_2_ profile during the rise and fall time, due to the position of the two sensors in the gas flow. The LGD module is positioned further in the stream of the gas sample (see Figure 2) and its gas cell needs to be filled and evacuated; hence the delay in the signal and the longer time constants. A higher flow rate drawn from the buffer tube would be able to compensate for these delays; however, the LGD module is already used at its maximum recommended flow rate. 

One participant shows almost constant CH_4_ concentration (close to the ~2 ppmv atmospheric level) (Figure 5a), while the other two present considerably higher CH_4_ concentration in exhaled breath, of ~6 ppmv and ~40 ppmv, respectively. 

### 3.3. Effects of Environmental Temperature

The influence of the temperature in the environment is also investigated by continuous measurements conducted at various temperatures. For this, the LGD module-based sensor is placed inside an environmental test chamber (MLR 351H, Sanyo Kogyo Co., LTD., Meguro-ku, Japan), where the temperature is set to 15 °C, 20 °C, 23 °C, 25 °C, 30 °C and 35 °C, respectively. Each temperature is maintained for approximately 30 min. The sensor is flushed with a continuous flow of ~50 ppmv CH_4_ prepared from a certified gas mixture of CH_4_ in synthetic air [101.5 (±2) ppmv, Linde Gas Benelux, Dieren, The Netherlands] and further diluted (50/50) by pure nitrogen. The relative humidity is controlled within the range of 6% to 29% during the experiment. Figure 6 shows the measurement values at different temperatures. Each value is obtained by averaging data over a 10-s period, while the standard deviations are calculated based on nine consecutive measurements. After applying the correction function, the measured values fell within the range of the applied concentration [50.75 (±1) ppmv, verified by the reference analyzer]. Despite variations in temperature, the largest deviation observed in the readings is 0.5 ppmv, representing a 1% deviation from the applied concentration. Therefore, these results indicate that with the correction function in place, the sensor exhibits good stability across various environmental temperatures.

### 3.4. Sensitivity and Long-Term Stability 

To assess the sensitivity and long-term stability of the setup, we conduct a series of measurements using a pure nitrogen sample and record the obtained values of the CH_4_ and CO_2_ from the LGD module-based sensor (after the correction). The associated Allan–Werle plots of these measurement series are presented in Figure 7 for both CH_4_ and CO_2_, fitted by a t^−1/2^ dependent line representative of the white noise contribution. An optimal averaging time of approximately 250 s results in a detection limit (precision) of ~5 ppbv for CH_4_ and ~80 ppmv for CO_2_. However, in practical applications, this may not be the most efficient option, since the measurement time becomes too long. For 10 s averaging the detection limits are ~30 ppbv and ~300 ppmv, and for a single-shot measurement (1.5 s) these values are ~80 ppbv and ~700 ppmv, for CH_4_ and CO_2_, respectively.

## 4. Conclusions and Outlook

We report on the development and the performance of a low-cost CH_4_/CO_2_ sensor for real-time breath analysis based on a commercial LGD module. The original module suffers from a significant influence of humidity on measuring both CH_4_ and CO_2_. Using an ultra-portable greenhouse gas analyzer (915-0011 Los Gatos Research, Mountain View, CA, USA) as a reference, correction functions are built into the data processing in order to account for these effects.

A compact breath sampler is also developed to be used together with the LGD module-based sensor for real-time breath measurements. Using a buffered volume approach, we show that the end-tidal portion of breath can be captured when the exhalation of the participant lasts for a minimum of five seconds; the limitation is due to the response time of the LGD module. The correction functions stay valid under different room temperatures of 15 °C to 35 °C, providing a good stability of the measurement. We obtain minimum detection limits (precision) of ~80 ppbv and ~700 ppmv for a single-shot measurement (1.5 s) and absolute minimum detection limits of ~5 ppbv and ~80 ppmv in ~250 s for CH_4_ and CO_2_, respectively. 

The results show a great promise of the developed system (after applying the correction functions) to be used as an inexpensive and portable (but still accurate and sensitive) device for on-line measurement of exhaled CH_4_. Compared to the competing technologies, the current sensor provides a very good balance of sensitivity, selectivity, real-time measurement, calibration-free operation, compact size and cost effectiveness. In addition, the capability of the sensor to simultaneously measure the concentration of CO_2_ eliminates the need for an independent CO_2_ sensor to guarantee the recording of the end-tidal part of the exhalation, further reducing the overall price, size and complexity, especially for in-situ operations. 

The recent improvements in the development of single-mode and widely tunable solid state lasers emitting in the mid-infrared (MIR) region, namely interband cascade lasers [31,32,33], promise more stable operation and lower cost. Since these lasers can operate at the strongest absorption band of CH_4_ (~3.3 µm), they are the most interesting candidate to replace the NIR tunable diode lasers in our sensor. The absorption features of CH_4_ at ~3.3 µm is almost two orders of magnitude stronger that its absorption features at ~1.6 µm. Therefore, despite the higher relative intensity noise of the interband cascade lasers and lower noise performance of the MIR photodetectors, a shorter interaction length is usually sufficient for the sensors based on interband cascade lasers to achieve the same detection sensitivity of NIR-based sensors. In recent years, sensitive CH_4_ sensors based on interband cascade lasers have been shown in laboratory-based [34] and transportable [35] setups, while compact designs are beginning to gain more attention [36]. 

On the high-end side, i.e., laboratory-based spectroscopy systems, mid-infrared optical frequency combs [37] show a great potential to be used for breath analysis. State-of-the-art systems can provide an ultra-broad spectral coverage with a very low intensity noise in the molecular fingerprint region (2–12 μm). Therefore, they can simultaneously detect multiple species with a very high sensitivity, either in a dual-comb spectroscopy scheme [38,39] or combined with a Fourier transform spectrometer [40]. Different proof-of-principle experiments have already demonstrated the capability of similar systems for breath analysis [41,42].

## Figures and Tables

**Figure 1 sensors-24-01077-f001:**
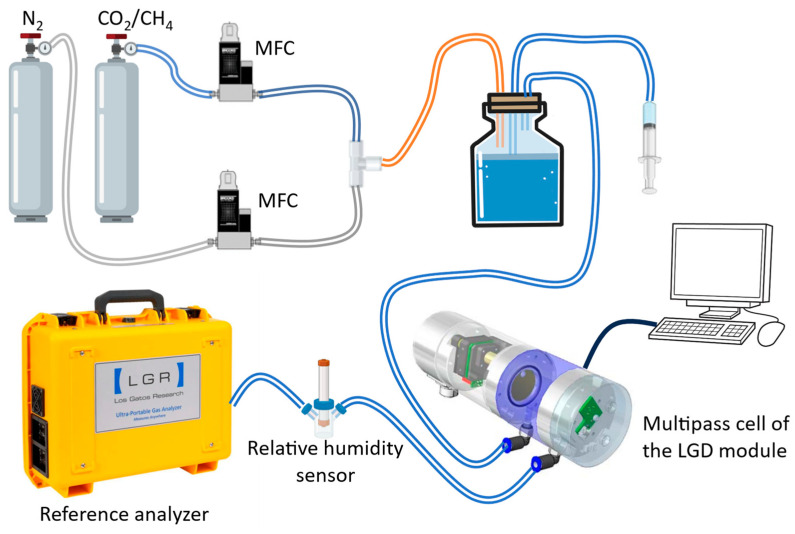
Schematic representation of the experimental setup to investigate the effects of humidity on the CH_4_/CO_2_ concentrations from the LGD module.

**Figure 2 sensors-24-01077-f002:**
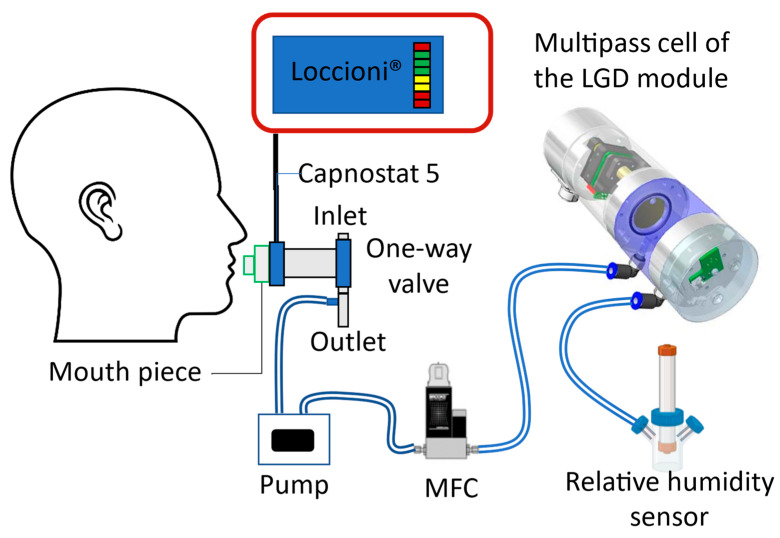
Schematic representation of the experimental setup for real-time breath measurement.

**Figure 3 sensors-24-01077-f003:**
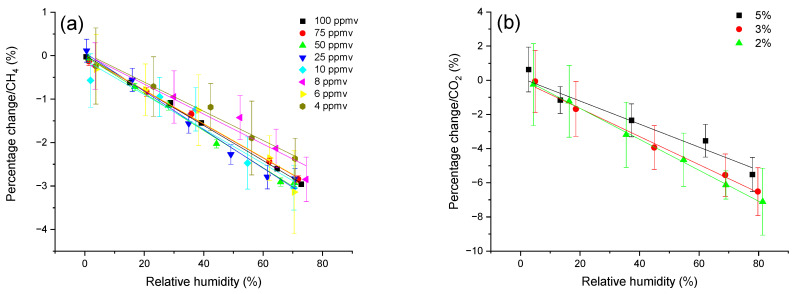
The percentage change in the measured concentrations of (**a**) CH_4_ and (**b**) CO_2_ using the LGD module in different relative humidity. The measured concentration values of both species show a relative (rather) linear trend of decreasing with respect to increase in the relative humidity from 0% to ~80%, around 3% for CH_4_ and 7% for CO_2_.

**Figure 4 sensors-24-01077-f004:**
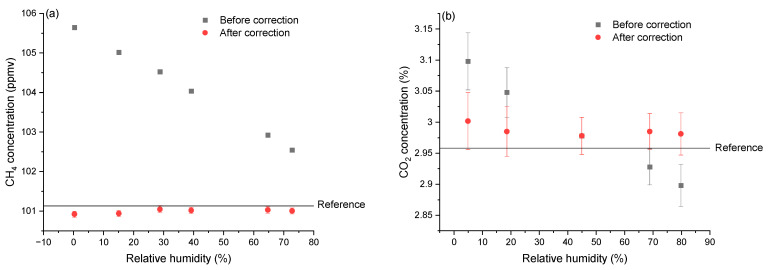
Comparison between values before and after the correction in various relative humidity levels, for (**a**) CH_4_ and (**b**) CO_2_. Black squares are the raw values from the LGD module before the correction. Red circles are the values after the correction. The black lines and the red area are the values and standard deviations of the references.

**Figure 5 sensors-24-01077-f005:**
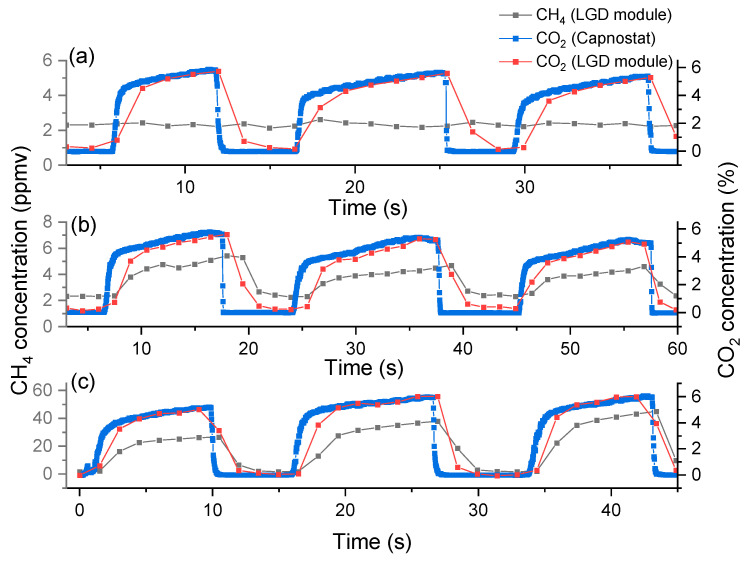
Real-time measurements of three breath profiles of CH_4_ (in black) and CO_2_ (in red) from three healthy participants (**a**–**c**) performed with the breath sensor based on the LGD module. For comparison, the CO_2_ profile recorded with the CAPNOSTAT^®^ 5 CO_2_ sensor is shown in blue.

**Figure 6 sensors-24-01077-f006:**
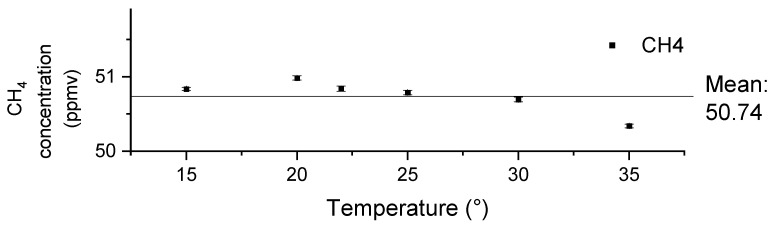
The measured values of CH_4_ after the correction in different environmental temperatures. Each value is obtained by averaging data over a 10-s period, the standard deviations are calculated based on nine consecutive measurements.

**Figure 7 sensors-24-01077-f007:**
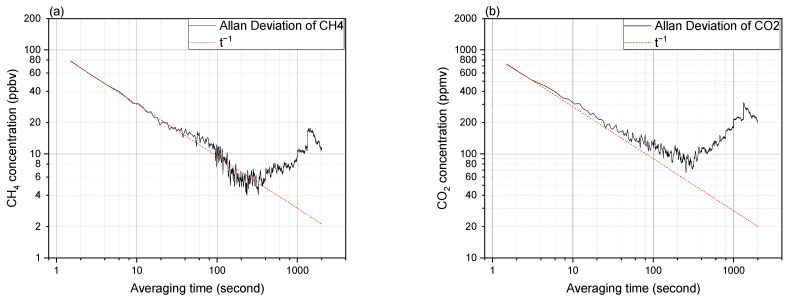
The Allan–Werle plots for detected concentration of (**a**) CH_4_ and (**b**) CO_2_. Fitted t^−1/2^ dependency representing the white noise contribution (red dashed line).

**Table 1 sensors-24-01077-t001:** Measurement errors before and after applying the correction.

Gas	Applied Concentration	Error before Correction	Error after Correction
CH_4_ (ppmv)	100	1.4~4.4%	<0.2%
	75	1.9~4.7%	<0.2%
	50	2.8~5.8%	<0.7%
	25	4.3~7.5%	<1.5%
	10	6.4~9.2%	<0.5%
	8	6.2~9.1%	<0.5%
	6	6.7~9.8%	<1%
	4	6.4~8.7%	<0.2%
	2	4.1~5.7%	<2.8%
CO_2_ (%)	2	1~3.4%	<1.4%
	3	0.6~4.7%	<1.4%
	5	0.1~3.9%	<1%

## Data Availability

Data is available upon request.

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
