# Peer review of "Real-Time Measurement of CH4 in Human Breath Using a Compact CH4/CO2 Sensor"

_sensors, 2024, doi:10.3390/s24041077_

Round 1

Reviewer 1 Report

Comments and Suggestions for Authors

The authors present a humidity correction for a commercial TDLS sensor for CO2/CH4. This particular type of sensor has found recently a wide uptake by the gas analysis community for several applications (e.g. monitoring emissions from livestock housings). Therefore, this work is timely. I recommend that the paper is accepted and only have some minor comments:

P4 “By using two mass flow controllers (SNM18204113B, Bronkhorst”. The number SNM18204113B seems not to refer to a certain model but more like a serial number.

 P5 “For the correction of the CO2 concentrations, we do not use the reference analyzer due to its limited guaranteed accuracy of 2%. Instead, we use a certified gas mixture of CO2 and synthetic air with 4.96% concentration” The certified gas mixture has a stated uncertainty of 2% and thus does not improve this. Please comment.

 P5 Figure 3a. For low CH4 concentrations (4-10 ppm which are probably diluted from the 10 ppm CH4) there seems to be a trend to less change for lower CH4 concentrations (e.g. for 4 ppm the change at a certain RH is smaller than for 10 ppm). This might be due to residual CH4 in the diluting gas (N2). Please state the impurity of CH4 in the N2 used as dilution gas as this may affect these lowest concentrations.

 P8 “50.75(± 1) ppmv” This and some other stated uncertainties in the documents seem to be too low. Next to the uncertainty of the gas standard, also e.g. the uncertainty contribution from the dilution needs to be included resulting in a higher uncertainty.

 P8 “After applying the correction function, the measured values fell within the range of the applied concentration [50.75(± 1) ppmv]. Despite variations in temperature, the largest deviation observed in the readings is 0.5 ppmv, representing a 1% deviation from the applied concentration. Therefore, these results indicate that with the correction function in place, the sensor exhibits good stability across various environmental temperatures.” For the applied relative humidity range of 6% - 29%, a maximum effect of 1% is expected based on Figure 3 and ignoring temperature. It would be good to show also the results without correction to see if the effect is much larger or not. It seems that the module is rather temperature insensitive by itself.

 P9 “We obtain minimum detection limits (precision) of ~80 ppbv and ~600 ppmv for a single-shot measurement (1.5 s)…” For CO2 the minimum detection limit seems higher (bit over 700 ppmv) than 600 ppmv looking to Fig 7.

Comments on the Quality of English Language

There are a few minor grammar errors in the manuscript which need to be corrected.

Reviewer 2 Report

Comments and Suggestions for Authors

1. The author should provide more details for the LGD module, such as the center wavelength , optical path length, and readout time. Without this information, it is not possible to determine the gas absorption intensity and therefore assess the detection performance of the instrument.

2. Is there any interference between CH4 and CO2? How do they affect the calibration of humidity? It is necessary to give the gas absorption wavelength.

3. Are the two gases detected using different wavelengths or the same wavelength with wavelength multiplexing or wavelength modulation techniques?

4. In section 3.4, "Sensitivity and long-term stability," what is the concentration of the gas measured over a long period of time?

Reviewer 3 Report

Comments and Suggestions for Authors

The authors report the performance improvement of a commercially available dual-gas sensor, after obtaining the correction function of correcting the effect from varying relative humidity, at the cost of extra monitoring its relative humidity. Further verification proves its validation in the case of a modest varying temperature.

However, as a research article, the novelty is insufficient. The results seem only apply to this commercial sensor from Axetris, i.e., LGD Compact-A CH4/CO2. If so, the results should be very helpful to update their products that can find extra market for moist applications.

1. The effect from the humidity remains unknown with no explanation. I guess it might come from the spectral overlap between the inevitable water and the target analyte. If so, is the deviation a reasonable value?

2. The working principle in not provided, as well as its laser wavelength, MPC type, optical length.

3. The novelty of this work has to be strengthened properly, especially its scientific contribution to this field.

Reviewer 4 Report

Comments and Suggestions for Authors

This manuscript reports a compact CH4/CO2 sensor for real-time measurement of CH4 in human breath. The highlights include 1) development of a compact and portable CH4 sensor that can also measure CO2, allowing for the extraction of end-tidal concentration of exhaled CH4; 2) identification and validation of correction functions for both CO2 and CH4 measurements to account for interferences with water and mutual interference between CH4/CO2 values; 3) development and testing of a custom-built breath sampler for real-time measurements of CH4 and CO2 in exhaled breath, demonstrating the capability to measure accurately exhaled CH4 and CO2 profiles. This manuscript is well organized, with solid data showing the sensor's temperature dependence, detection sensitivity, and long-term stability, providing a comprehensive understanding of its capabilities. Therefore, I think the paper should be accepted.

 Meanwhile, I have a few concerns that may further improve the manuscript:

1) The specific NIR wavelength used in the sensor module is not specified by the vendor, which limits the ability to fully understand the spectral interference of CH4 with water vapor in breath analysis.

2) The study focuses on developing and validating correction functions for CO2 and CH4 measurements. Still, it does not extensively explore other potential interferences or factors affecting the sensor's accuracy.

3) The study does not compare with other existing CH4 sensors or methods, limiting the ability to assess the novelty and superiority of the developed sensor.

4) The authors should add a few sentences discussing the future improvements, such as improving the sensor's sensitivity, e.g., by accessing a wavelength region with stronger absorptions, and extending the number of gas species by using optical combs (Nat. Commun. 14, 5037 (2023); Nat Commun 13, 2181 (2022); Nat. Commun. 11, 4164 (2020)).

Reviewer 5 Report

Comments and Suggestions for Authors

The article titled "Real-time measurement of CH4 in human breath using a com- 2 pact CH4/CO2 sensor" is very well written and can be accepted for publication in present form. 

Author Response

We thank the reviewer for the careful reading of the manuscript and the positive evaluation.

Round 2

Reviewer 2 Report

Comments and Suggestions for Authors

The manuscript has been done well. There is no further recommendation.

Author Response

We thank the reviewer for the comments.

Reviewer 3 Report

Comments and Suggestions for Authors

My biggest concern was the insufficient/ambiguous novelty. It’s good that the authors have emphasized it by extending the conclusion. Again, the authors claim that the corrections obtained in this work can improve its application to measure exhaled CH4 using a commercially available sensor, but without knowing its working principle or other details mentioned in my first comment.

Note that “Our breath sensor is****in the NIR region” is not working principle. Absorption corresponds to a lot of working principle/techniques, for example, spectroscopic ones, TDLAS, WMS, photoacoustic, etc. Currently, it remains unclear, let alone other details. Hence, it is impossible to judge its scientific contribution from the perspective of developing a sensor, or from the perspective of application in exhalation diagnostic. If the authors still don’t know the detailed information, I suggest that authors discuss the possibility of applying their “corrections” to other sensors of different working principle/techniques, or it only applies to this specific OEM module.

In breath diagnostic, usually concentration information of different gas species is needed. It’s valuable to extend the gas species to be analyzed. Hence, the 4th comment of Reviewer 4, “extending the number of gas species by using optical combs…” is reasonable. However, the authors haven’t replied it.
